# Stable Fault Tolerant Controller Design for Takagi–Sugeno Fuzzy Model-Based Control Systems via Linear Matrix Inequalities: Three Conical Tank Case Study

**Himanshukumar R. Patel** *  **and Vipul A. Shah** †

Department of Instrumentation and Control, Faculty of Technology, Dharmsinh Desai University, Gujarat, 387001 Nadiad, India; vashahin2010@gmail.com

* Correspondence: himanshupatelp32@gmail.com or himanshupatel.ic@ddu.ac.in; Tel.: +91-76-983-08667
† These authors contributed to this work as a Ph.D guide.

**Abstract:** This paper deals with a methodical design approach of fault-tolerant controller that gives assurance for the the stabilization and acceptable control performance of the nonlinear systems which can be described by Takagi–Sugeno (T–S) fuzzy models. Takagi–Sugeno fuzzy model gives a unique edge that allows us to apply the traditional linear system theory for the investigation and blend of nonlinear systems by linear models in a different state space region. The overall fuzzy model of the nonlinear system is obtained by fuzzy combination of the all linear models. After that, based on this linear model, we employ parallel distributed compensation for designing linear controllers for each linear model. Also this paper reports of the T–S fuzzy system with less conservative stabilization condition which gives decent performance. However, the controller synthesis for nonlinear systems described by the T–S fuzzy model is a complicated task, which can be reduced to convex problems linking with linear matrix inequalities (LMIs). Further sufficient conservative stabilization conditions are represented by a set of LMIs for the Takagi–Sugeno fuzzy control systems, which can be solved by using MATLAB software. Two-rule T–S fuzzy model is used to describe the nonlinear system and this system demonstrated with proposed fault-tolerant control scheme. The proposed fault-tolerant controller implemented and validated on three interconnected conical tank system with two constraints in terms of faults, one issed to build the actuator and sond is system component (leak) respectively. The MATLAB Simulink platform with linear fuzzy models and an LMI Toolbox was used to solve the LMIs and determine the controller gains subject to the proposed design approach.

**Keywords:** actuator fault; fuzzy control; linear matrix inequalities; T–S model-based fuzzy control; parallel distributed compensation; stability condition; system component fault; three conical tank; Takagi–Sugeno fuzzy model

## 1. Introduction

Many years ago, in 1965, fuzzy sets and logic system were introduced by renowned researcher Zadeh. Although fuzzy logic is applied in numerous complex industrial applications, for example steam engines by Mamdani's, speed control of a DC motor applications and boiler fusion [1,2], Kickert's proposed linguistic rules that describe human operator's control strategy which is applied to control warm water plant [3], and Ostergaard's introduced fuzzy logic control of heat exchange system [4], fuzzy sets and control theory have ability to replicate operator's control strategy in to linguistic rules however stability analysis, robustness and optimality features are not existed, contradictory modern

and conventional control theories have these features which are very significant in fail-safe critical engineering applications. Therefore, from long time being considerable attentiveness is gaining in terms of stability evaluation and methodical design of fuzzy control systems which are inherently nonlinear.

Describing linear systems from nonlinear systems by the "Takagi–Sugeno (T–S) fuzzy model", initially introduced by Takagi and Sugeno [5], which works on the local dynamics of the nonlinear system, the linear models described by different state space regions, particularly at an operating region of the nonlinear system. The overall system is obtained by fuzzy blending of these linear models [2]. The design procedure of the "T–S fuzzy model-based controller" is explained in detail by [6,7], the author of [6] explain the systematic approach to design state feedback fuzzy controllers for T–S fuzzy models and in [7], the authors implement real-time T–S fuzzy model-based controller for shunt compensator. Based on the T–S fuzzy model, Wang et al. designed a linear controller by using "parallel distributed compensation (PDC)", the system was described by T–S fuzzy model, in this scheme a linear state feedback controller was designed for each linear model [2,8]. Since way back in 1992, Tanaka and Sugeno was proposed sufficient stabilization conditions for T–S fuzzy controllers [9]. However,form these conditions the existence of positive definite matrix $P$ is necessitate which satisfies a set of Lyapunov inequalities [2]. The authors of [2] proposed an overall fuzzy model by fuzzy blending of all liner model at particular operating region, the controller synthesis is a difficult task for nonlinear system which described by T–S fuzzy model, this problem is turned into simple mathematical equation solver by involving "linear matrix inequalities (LMIs)". These LMIs can be solved using optimization methods like the interior-point convex optimization method [2,10]. The less conservative stabilization and optimum performance sufficient conditions for T–S fuzzy control systems can be transformed into set of LMIs, LMI Toolbox can be used to solve these LMIs [11]. Finally, Khaber et. al. proposed a T–S fuzzy model for inverted pendulum on a moving cart which is highly nonlinear system, thereafter author proposed state feedback controller for two T–S fuzzy-rule and MATLAB software ware used to wrote the code and solve LMIs [12]. However, it is crucial to identify the common Lyapunov function that satisfies sufficient stability conditions of all fuzzy models for complex nonlinear systems. Beyond this limitation, a mathematical model of the nonlinear system is a key challenges [13].

Controlling of process variable like flow rate and liquid level in closed tanks are the very common control problems in the chemical process, food processing, cement industries and petrochemical industries [14–17]. Generally, the liquids are pumped and stored in the tanks for processing; again it is pumped to other tanks for other operations. The conical tanks are widely used in liquid treatment industry, concrete industry and hydro metallurgical industries [18]. In the present decade, researchers focusing on the level control problem and numerous literature exist that covers conventional to state of the art level control of cylindrical and cone-shaped tanks. This literature addressed and validated control schemes by simulation as well as experimental setup and several control techniques have been employed [18–22]. The three interacting conical tank level (TICTL) process is a typical two input two output (TITO) process which exhibits nonlinear characteristics and dynamic coupling effect between inputs and outputs. The control of TITO process requires dedicated multiloop or multivariable control system. Commonly, the process industries employ multiloop proportional integral derivative (PID) controller because of its simple structure, robustness and failure tolerance [23]. The multiloop PID controllers produce better control performance for the system with modest interaction. But it fails to provide desirable control performance for the system with severe interaction effect between inputs and outputs.

The decoupled control scheme and multivariable centralized controller are designed with linear PI/PID controller, which are failed to produce reasonable performance for the nonlinear system. Hence, the adaptive PI/PID control has been designed for nonlinear multi input multi output (MIMO) systems by combining family of linear PI/PID controller with gain scheduling scheme. The adaptive PI/PID controller with gain scheduling scheme is limited up to abrupt changes of the process operating points across the boundaries of the operating regions, and it will fail at points other than these operating

points. To overcome this limitation author of [24] have demonstrated the fuzzy gain scheduler-based PID control for process control application. In recently author of the [25] has been claimed that fuzzy logic based gain scheduler provides satisfactory control performance for nonlinear system. In article [26], the author used multimodal-based gain scheduler for controlling the level of liquid in the single conical tank (SISO) process, where the linear PID controllers are found in three operating regions and then weighted scheduler is designed to adjust the Controller parameter based on the operating regions. Reference [27] has developed a fuzzy logic controller for nonlinear spherical tank level process, in that fuzzy rule base is tuned using a genetic algorithm and it has been claimed that fuzzy logic control is more remarkable than conventional PI control. References [28–30] have demonstrated fuzzy gain scheduler-based PID controller for nonlinear MIMO process. After the motivation and literature survey for T–S fuzzy model-based controller and fuzzy logic base controller for nonlinear system, in this article we design a stable fault tolerant controller for the T–S fuzzy model-based control system using LMIs approach. To validate the proposed fault tolerant approach we take TICTL process as a case study. In artificial intelligence (AI) neural network (NN) is also used to fault tolerant control applications, NN will used to estimate the faults and take controlling action accordingly, some application and implementation of NN is presented in [17,31]. The author of [32] design stable T–S fuzzy controller by optimization, and in [33] author has design linear controller by applying Takagi–Sugeno fuzzy model and adding partial uncertainty in reference trajectory.

The major contributions of the paper are as follows. A novel stable fault tolerant controller is designed based on T–S fuzzy model and controller synthesis is done via LMIs for the three conical tank level control process and considering model uncertainties and two type of faults. The state feedback control law is designed based on T–S fuzzy model and parallel distribution compensation (PDC) method. The quadratic stability theory is used to prove the quadratic stability of the uncertain three conical tank system based on the T–S fuzzy model. We provide an effective method of designing fuzzy multigain controllers according to PDC algorithm and linear matrix inequality (LMI) to ensure the controller satisfies fuzzy controller performance.

The objective of this work is to develop a T–S fuzzy model for the proposed highly nonlinear TICTL process and then to design stable fault tolerant control system to control the liquid level of tank 1, tank 3 irrespective of fault occurs into the system. In this paper, parallel distributed compensation is used to design the linear controller for each linear model. The controller performance are demonstrated using three error indices IAE, ISE and IATE. The state feedback gains are found for all linear T–S fuzzy model in region 3 of TICTL process hence it gives guarantee stability and optimum performance. Also it overcame the interaction effect between loops and improved the servo, regulatory performance of closed loop system.

This work is organized as follows: Section 2 includes process description of TICTL process, mathematical model of the process along with linerized model with specific region 3. Section 3 presents the Fault tolerant controller using T–S fuzzy model-based controller for TICTL process subject to two possible faults. Section 4 presents the simulation results of regulatory and servo response of TICTL process with and without faults, along with error calculation is presented. Finally in Section 5 the main conclusions of the work and future work are drawn.

## 2. Three Conical Tank Process Description

The TICTL process prototype model is depicted in Figure 1. The proposed level control system made up of three interconnected cone- shaped tanks connected by cylindrical pipe. The system consist two input flow rate $F_{in1}$ and $F_{in2}$ at tank 1 and tank 3, three outlet flow rate $F_1$, $F_2$, and $F_3$ for tank 1, tank 2 and tank 3 respectively. Also, systems having interaction using interconnected pipes between tank 1–2 and tank 3–2, $F_{12}$ is a flow rate of interaction between tank 1–2 and $F_{32}$ is a flow rate of interaction between tank 3–2. The gate values $V_1$, $V_2$, $V_3$, and interaction valve $V_{12}$ and $V_{32}$ are partially on and kept at constant position. The interaction effect due to two level values $h_1 - h_2$ and $h_3 - h_2$ of process, and it can be changed by the valve position $V_{12}$ and $V_{32}$. The outer two conical tanks received

inflow of liquid from variable speed pumps. The manipulated inputs of system are the voltage applied to the pumps. The range of input voltage is 0 to 5 V, which is directly proportional to derivative of inflow with respect to time. The differential pressure type (DP) transmitter used for measuring the level in the conical tank in terms of milliampere (mA). The prime objective of this system is to maintain the liquid height in the conical tank 1 and tank 3 by changing the applied input voltages to Pump 1 and 2. In the prototype system of (TICTL) two possible faults are considered for validation of proposed controller, one is system component ($f_{sys}$) (leak) fault in tank 1 and tank 3 other is actuator ($f_a$) fault in $CV_1$ and $CV_2$.

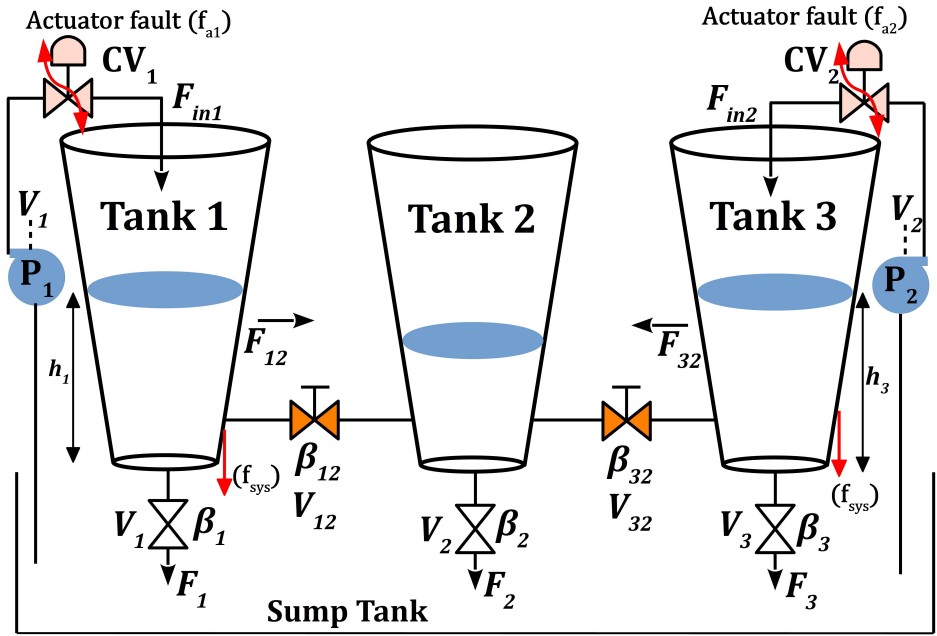

**Figure 1.** Prototype schematic of three interacting conical tank level (TICTL) process.

### 2.1. Mathematical Modeling of TICTL Process

The mathematical model of TICFTL process is derived from the mass balance equation. The single conical tank system shown in Figure 2.

The mathematical model for single frustum conical tank process is derived using the conservation of mass and Bernoulli's principle as follows,

Rate of accumulation = Rate of inflow − Rate of Outflow ([17])

$$\frac{\mathrm{d}Vol}{\mathrm{d}t} = F_{in} - F_{out}, \tag{1}$$

where *Vol* is a volume of liquid in the conical tank. The volume of liquid change due its varying surface area of the tank. The Volume of conical tank Volume is

$$Vol = \frac{\pi}{3}\left(r_b^2 + r^2 + r_{br}\right), \tag{2}$$

where $r_b$ is the bottom radius of tank, and *r* is the top radius of liquid. The varying top radius of liquid level is found using trigonometric law.

$$\tan\theta = \frac{NM}{XN} = \frac{YZ}{XY}, \tag{3}$$

where $\theta$ is the angle of frustum conical slope. NM = $r_s$, is an incremental radius of liquid level due to slope surface.

$$r_s = \frac{R_s}{H}h = \frac{(R - r_b)}{H}h. \tag{4}$$

Top radius of liquid level, $r = r_b + r_s$,

$$r = r_b + \frac{(R - r_b)}{H}h. \tag{5}$$

After substituting $r$ value in Equation (2), the volume of liquid in conical tank becomes,

$$Vol = \frac{\pi}{3}\left[3r_b^2 h + 3r_b\left(\frac{R - r_b}{H}\right)h^2 + \left(\frac{R - r_b}{H}\right)^2 h^3\right]. \tag{6}$$

After substituting Equation (5) into Equation (1),

$$\frac{dh}{dt} = \frac{F_{in} - \beta a\sqrt{2gh}}{\frac{\pi}{3}\left[3r_b^2 + 6r_b\left(\frac{R - r_b}{H}\right)h + 3\left(\frac{R - r_b}{H}\right)^2 h^2\right]} \tag{7}$$

where $F_{out} = \beta a\sqrt{2gh}$ $a$ is a cross section area of outlet pipe and $\beta$ is the ratio of gate valve opening ($\beta$ varies from 0 to 1). When the valve is fully closed, $\beta$ is 0, and when the valve is fully open $\beta$ is 1. V is input voltage, $v$ is the pump gain. Similarly, the mathematical model for the TICTL process is developed and given by following equations,

$$\frac{dh_1}{dt} = \frac{v_1 V_1 - \beta_1 a_1 \sqrt{2gh_1} - sign(h_1 - h_2)\beta_{12}a_{12}\sqrt{2g|h_1 - h_2|}}{\frac{\pi}{3}\left[3r_{b1}^2 + 6r_{b1}\left(\frac{R - r_{b1}}{H_1}\right)h_1 + 3\left(\frac{R - r_{b1}}{H_1}\right)^2 h_1^2\right]} \tag{8}$$

$$\frac{dh_2}{dt} = \frac{sign(h_3 - h_2)\beta_{32}a_{32}\sqrt{2g|h_3 - h_2|} + sign(h_1 - h_2)\beta_{12}a_{12}\sqrt{2g|h_1 - h_2|} - \beta_2 a_2\sqrt{2gh_2}}{\frac{\pi}{3}\left[3r_{b2}^2 + 6r_{b2}\left(\frac{R - r_{b2}}{H_2}\right)h_2 + 3\left(\frac{R - r_{b2}}{H_2}\right)^2 h_2^2\right]} \tag{9}$$

$$\frac{dh_3}{dt} = \frac{v_2 V_2 - \beta_3 a_3 \sqrt{2gh_3} - sign(h_3 - h_2)\beta_{32}a_{32}\sqrt{2g|h_3 - h_2|}}{\frac{\pi}{3}\left[3r_{b3}^2 + 6r_{b3}\left(\frac{R - r_{b3}}{H_3}\right)h_3 + 3\left(\frac{R - r_{b3}}{H_3}\right)^2 h_3^2\right]}. \tag{10}$$

The TICTL process parameters are given in Table 1. The flow rates are a function of applied input voltage. The TICTL process exhibits nonlinear characteristics, hence the operating regions are found using a piecewise linearization method for controller design. The operating points obtained from input–output characteristic and three region is identified which is tabulated in the Table 2.

**Table 1.** Parameter for three interacting conical tank level (TICTL).

| Sr. No. | Parameters | Value with Unit |
|:---:|:---:|:---:|
| 1 | Top radius of conical tank $R_1$, $R_2$ and $R_3$ | 20 cm |
| 2 | Bottom radius of conical tank $r_b$ | 14 cm |
| 3 | Height of conical tank $H_1$, $H_2$ and $H_3$ | 55 cm |
| 4 | Pump 1, 2 gain $v_1$ and $v_2$ | 25 cm$^3$/V·s |
| 5 | Voltage applied to the Pump 1,2 $V_1$ and $V_2$ | (0–5) V |
| 6 | Valve coefficient $\beta_1$ | 0.33 |
| 7 | Valve coefficient $\beta_2$ | 0.33 |
| 8 | Valve coefficient $\beta_3$ | 0.33 |
| 9 | Valve coefficient $\beta_{12}$ | 0.2 |
| 10 | Valve coefficient $\beta_{32}$ | 0.2 |

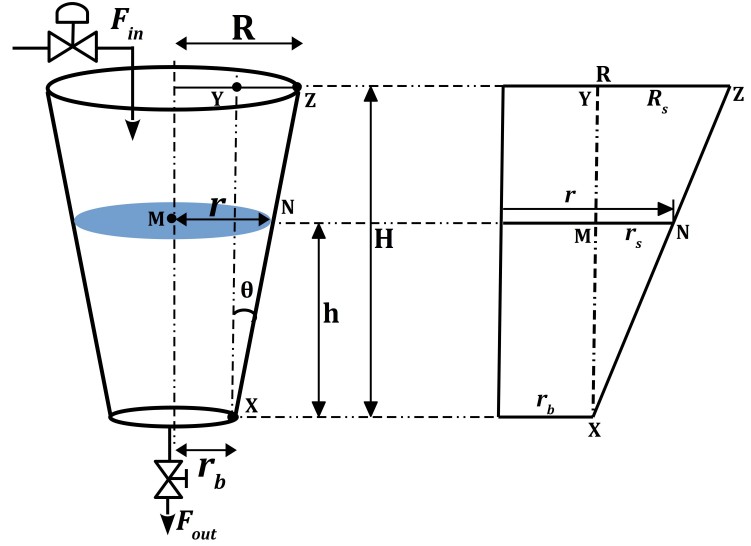

**Figure 2.** Volume of liquid in the conical tank.

**Table 2.** Operating point conditions for TICTL.

| Region | Operating Point Tank 1 | | Operating Point Tank 1 | |
|--------|------------------------|---|------------------------|---|
| | $F_{in1}$ | $h_1$ | $F_{in2}$ | $h_2$ |
| Region 1 | (0–1.5) V | (0–4.76) cm | (0–1.5) V | (0–4.76) cm |
| Region 2 | (1.5–3) V | (4.58–12.14) cm | (1.5–3) V | (4.58–12.14) cm |
| Region 3 | (3–5) V | (12.14–28.91) cm | (3–5) V | (12.14–28.91) cm |

## 2.2. Linearization of TICTL Model

The state space model and transfer function model is obtained around the operating points using Jacobian linearization. State equation is described as presented in [20],

$$
\begin{cases}
\dot{X} = Ax + Bu \\
Y = Cx + Du
\end{cases}
\tag{11}
$$

where X is the states of the process $[h_1, h_3]$ and u is the input vector of process $[V_1, V_2]$. The A, B matrices are the state matrix and input matrix of the state space model.

$$
\begin{bmatrix} \frac{dh_1}{dt} \\ \frac{dh_2}{dt} \\ \frac{dh_3}{dt} \end{bmatrix} =
\begin{bmatrix} \frac{\partial f_1}{\partial h_1} & \frac{\partial f_1}{\partial h_2} & \frac{\partial f_1}{\partial h_3} \\ \frac{\partial f_2}{\partial h_1} & \frac{\partial f_2}{\partial h_2} & \frac{\partial f_2}{\partial h_3} \\ \frac{\partial f_3}{\partial h_1} & \frac{\partial f_3}{\partial h_2} & \frac{\partial f_3}{\partial h_3} \end{bmatrix}
\begin{bmatrix} h_1 \\ h_2 \\ h_3 \end{bmatrix} +
\begin{bmatrix} \frac{\partial f_1}{\partial V_1} & \frac{\partial f_1}{\partial V_2} \\ \frac{\partial f_2}{\partial V_1} & \frac{\partial f_2}{\partial V_2} \\ \frac{\partial f_3}{\partial V_1} & \frac{\partial f_3}{\partial V_2} \end{bmatrix}
\begin{bmatrix} V_1 \\ V_2 \end{bmatrix}
\tag{12}
$$

$$
\begin{bmatrix} Y_1 \\ Y_2 \\ Y_3 \end{bmatrix} =
\begin{bmatrix} 1 & 0 & 0 \\ 0 & 0 & 1 \end{bmatrix}
\begin{bmatrix} h_1 \\ h_2 \\ h_3 \end{bmatrix} +
\begin{bmatrix} 0 & 0 \\ 0 & 0 \\ 0 & 0 \end{bmatrix}
\begin{bmatrix} V_1 \\ V_2 \end{bmatrix}
\tag{13}
$$

where $f_1$ is the function $\frac{dh_1}{dt}$, $f_2$ is the function $\frac{dh_2}{dt}$, $f_3$ is the function $\frac{dh_3}{dt}$. Y is the output vector $[h_1, 0, h_3]$, C is the output matrix, D is the feed-forward input matrix. $Y_1$, $Y_2$ are the outputs of the TICTL process.

The linear state space model for fault-free TICTL system is found using Equation (12) and A, B, and C matrices are found as follows:

$$
A = \begin{bmatrix}
-\dfrac{b_{12}}{\sqrt{h_1 - h_2}} - \dfrac{b_1}{\sqrt{h_1}} & -\dfrac{b_{12}}{\sqrt{h_1 - h_2}} & 0 \\
-\dfrac{b_{12}}{\sqrt{h_1 - h_2}} & \dfrac{b_{32}}{\sqrt{h_3 - h_2}} - \dfrac{b_{12}}{\sqrt{h_1 - h_2}} - \dfrac{b_2}{\sqrt{h_2}} & \dfrac{b_{32}}{\sqrt{h_3 - h_2}} \\
0 & \dfrac{b_{32}}{\sqrt{h_3 - h_2}} & -\dfrac{b_{32}}{\sqrt{h_3 - h_2}} - \dfrac{b_3}{\sqrt{h_3}}
\end{bmatrix}
$$

$$B = \begin{bmatrix} \frac{1}{G} & 0 \\ 0 & 0 \\ 0 & \frac{1}{G} \end{bmatrix}, \quad C = \begin{bmatrix} 1 & 0 & 0 \\ 0 & 0 & 1 \end{bmatrix}, \tag{14}$$

where $G = \frac{\pi}{3}\left[3r_b^2 + 6r_b\left(\frac{R-r_b}{H}\right)h + 3\left(\frac{R-r_b}{H}\right)^2 h^2\right]$, $b_1 = \frac{\beta_1 a_1 \sqrt{2g}}{2G}$, $b_2 = \frac{\beta_2 a_2 \sqrt{2g}}{2G}$, $b_3 = \frac{\beta_3 a_3 \sqrt{2g}}{2G}$, $b_{12} = \frac{\beta_{12} a_{12} \sqrt{2g}}{2G}$ and $b_{32} = \frac{\beta_{32} a_{32} \sqrt{2g}}{2G}$.

## 3. Fault Tolerant Controller Design

In this article, the T–S fuzzy model-based controller is designed for controlling the TICTL process with and without two possible faults (i.e., system component (leak) and Actuator Fault) into the system. In order to preserve the system stability and optimum control performance of the nonlinear system, first T–S fuzzy model approach is adopted to select appropriate linear model of the TICTL process within the operating region 3 given in Table 2. Once the appropriate linear model is selected then linear controllers are designed for each linear model around at operating region using PDC. The sufficient stabilization and optimum performance conditions for T–S fuzzy control systems can be represented by a set of LMIs which can be solved using software packages such as MATLAB's LMI Toolbox. The step-by-step procedure is given in following subsection.

### 3.1. Preliminaries and Takagi–Sugeno Fuzzy Mode

One of the key advantage of the T–S fuzzy model is to present nonlinear systems in linear fuzzy models at a local operating region by IF–THEN rules, where each IF–THEN rules indicates the local dynamics of the nonlinear system by a linear system [2]. Then by combining all the possible fuzzy linear models at a local operating region, the overall fuzzy model is obtained for a nonlinear system. The $i^{th}$ fuzzy rule of a continuous T–S fuzzy system can be written as:

IF $z_1(t) = M_{i1} \cdots$ and $z_n(t) = M_{in}$ THEN $\dot{X} = A_i x + B_i u$ where $z_1(t), z_2(t), \cdots, z_n(t)$ are premise variables, $x \in \mathbb{R}^{n \times 1}$ is the state vector, $r$ is the number of rules, $M_{ij}$ are input fuzzy sets, $u \in \mathbb{R}^{m \times 1}$ is the input, $A_i \in \mathbb{R}^{n \times n}$ is the system state matrix and $B_i \in \mathbb{R}^{n \times n}$ is the input matrix. $z_1(t), z_2(t), \cdots, z_n(t)$ premise variables should be functions of the state variables, external disturbances and/or time. Given a pair of $(x(t), u(t))$ the final output of the fuzzy system is inferred as:

$$\dot{x} = \frac{\sum_{i=1}^{r} w_i(z)(A_i x + B_i u)}{\sum_{i=1}^{r} w_i(z)} \tag{15}$$

where $z = [z_1(t), z_2(t), \cdots, z_n(t)]$ and

$$w_i(z) = \prod_{j=1}^{n} M_{ij}(z_j) \tag{16}$$

$M_{ij}$ is the membership function of the $j$th fuzzy set in the $i$th fuzzy rule. Let

$$a_i(z) = \frac{w_i(z)}{\sum_{i=1}^{r} w_i(z)}. \tag{17}$$

Then (1) may be expressed as

$$\dot{x} = \sum_{i=1}^{r} a_i(z)(A_i x + B_i u). \tag{18}$$

Since $w_i(z) \geq 0$ and $\sum_{i=1}^{r} w_i(z) > 0$ we have $\sum_{i=1}^{r} a_i(z) = 1$ and $a_i(z) \geq 0$.

### 3.1.1. Parallel Distributed Compensation

PDC contributes an efficient and effortless methodology, which designs a fuzzy controller for the control of a nonlinear system described by a T–S fuzzy model in a different operating region. In PDC design, a set of control rules is fabricated for a set of the T–S fuzzy model. The designed fuzzy controller by PDC and the T–S fuzzy model have the same fuzzy sets in antecedent parts. Linear controllers exist in consequent or concluding part of the control rules. Thus, the *i*-th rule of the controller is as follows:

IF $z_1(t) = M_{i1} \cdots$ and $z_n(t) = M_{in}$ THEN $u = -K_i x$. In this expression, fuzzy control rules have state feedback controllers in consequent or concluding parts. The principle structure of the PDC is presented in Figure 3.

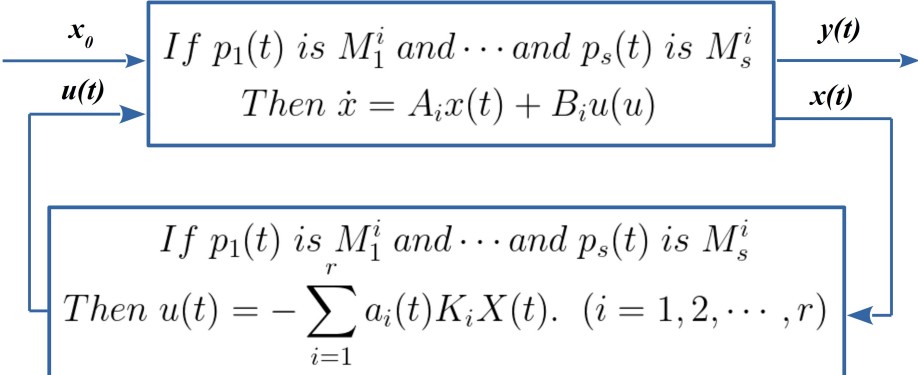

**Figure 3.** General diagram of parallel distributed compensation.

The overall fuzzy controller is described mathematically as

$$u = -\frac{\sum_{i=1}^{r} w_i(z) K_i x}{\sum_{i=1}^{r} w_i(z)} = -\sum_{i=1}^{r} a_i(z) K_i x. \tag{19}$$

The main design problem is to find out the local feedback gains $K_i$ in consequent parts. Note that the overall fuzzy controller (it is a combination of all linear model at particular operating range of nonlinear system) is nonlinear in general.

A linear state feedback controller for each linear subsystem is designed using the PDC algorithm. The basic idea of the PDC algorithm is to design a controller corresponding to each rule of the fuzzy model described the control object. The designed controller is nonlinear, composed of multiple linear controllers. The control principle of the PDC algorithm design controller is illustrated in Figure 4. Although the controller of the whole system is designed based on each local linear subsystem, according to the Lyapunov method, the whole system has been proved to be globally asymptotically stable [34].

### 3.1.2. Stability Analysis of Fuzzy Control Systems and Controller Synthesis Using LMIs

Let us replace (5) in (4). Then the closed loop control system is expressed as [2]

$$x = \sum_{i=1}^{r} \sum_{j=1}^{r} a_i(z) a_j(z) (A_i - B_i K_j) x \tag{20}$$

Closed loop system (6) can be written in [2] which is presented as

$$\dot{x} = \sum_{i=1}^{r} a_i(z) a_i(z) (A_i - B_i - K_i) + \sum_{i=1}^{r} \sum_{j=1}^{r} a_i(z) a_i(z) G_{ij}. \tag{21}$$

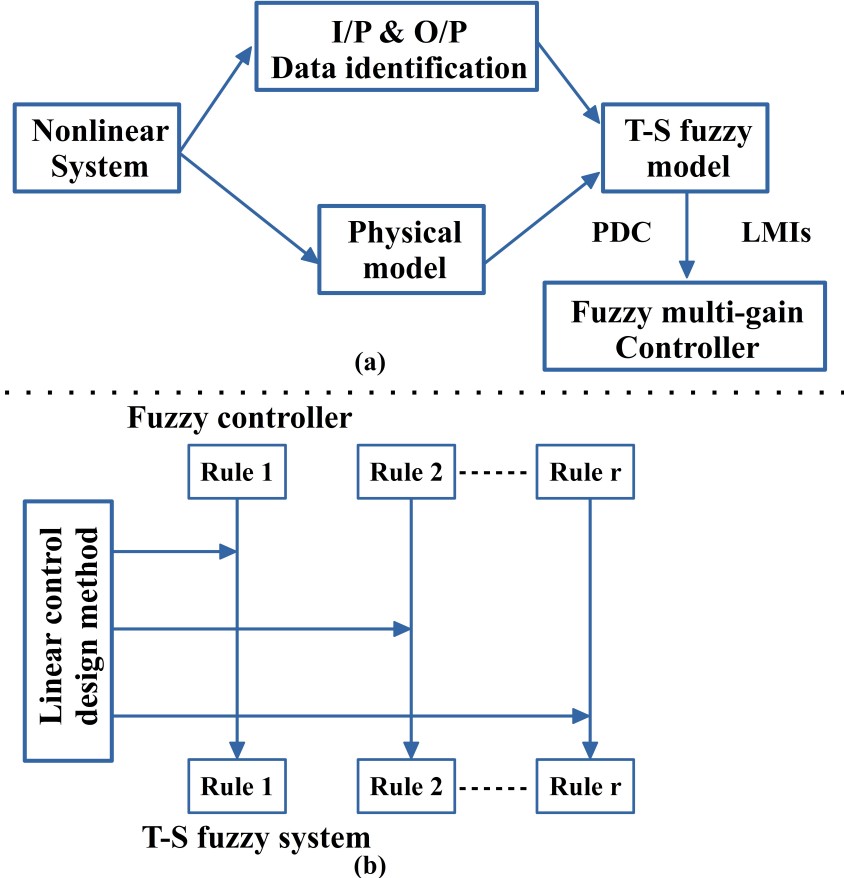

**Figure 4.** Systematic procedure diagram to design stable Fault tolerant controller. (**a**) Design flow chart of the fuzzy controllers. (**b**) Parallel distributed compensation (PDC) algorithm diagram.

where

$$\begin{cases} \boldsymbol{G_{ij}} = \left(A_i - B_i K_j\right) + \left(A_j - B_j K_i\right) \\ i < j \text{ subject to } \boldsymbol{a_i}\left(\boldsymbol{z}\right) \times \boldsymbol{a_j}\left(\boldsymbol{z}\right) \neq \phi \end{cases} \tag{22}$$

**Theorem 1** ([8]). *The closed loop fuzzy control system (6) is globally asymptotically stable if there exists a common positive-definite matrix P which satisfies the following Lyapunov inequalities:*

$$\begin{cases} \left(A_i - B_i K_i\right)^T P + P \left(A_i - B_i K_i\right)^T < 0, \ i = 1, 2, \cdots, r \\ \boldsymbol{G_{ij}}^T P + P \boldsymbol{G_{ij}} < 0, \ i < j \leq r \end{cases} \tag{23}$$

*Pre-multiplying and post-multiplying the both sides of inequalities in (9) by $P^{-1}$ and using the following change of variables:*

$$\begin{cases} Y = P^{-1} \\ X_i = K_i Y \end{cases} \tag{24}$$

*we obtain the following LMIs:*

$$\begin{cases} Y A_i^T + A_i Y - B_i X_i - X_i^T B_i^T < 0 \\ Y \left(A_i + A_j\right)^T + \left(A_i + A_j\right) Y - L_{ij} - L_{ij}^T < 0 \end{cases} \tag{25}$$

*where $L_{ij} = B_i X_j + B_j X_i$.*

*If the LMIs shown in (11) have a common positive definite solution then the stability is guaranteed [2].*

### 3.2. T–S Fuzzy Model of the TICTL

Three inputs used for the Takagi-Sugeno model are the conical tank 1 height $h_1$, conical tank 2 height $h_2$ and the pump voltage. The three inputs are fuzzified using two fuzzy sets B, S whose membership functions ($\mu$) are presented in Figure 5. For the positive set, P

$$\mu_P(x_i) = \begin{cases} 0, & x_i < -B_1 \\ \frac{x_i+B}{2B}, & -B_1 \le x_i \le +B_1 \\ 1, & x_i < +B_1 \end{cases} \tag{26}$$

$$\mu_n(x_i) = \begin{cases} 0, & x_i < -B_2 \\ \frac{x_i+B}{2B}, & -B_2 \le x_i \le +B_2 \\ 1, & x_i < +B_2 \end{cases}, \tag{27}$$

where $x_i$ stands for input to the fuzzy controller at the $k_{th}$ sampling instant. For the small set S, it may be written as in Equation (2). For two sets membership function are depicted in Figure 5 with $B_1$, $B_2$, $B_3$ and $B_4$ denoting the bounds. Using the Takagi–Sugeno fuzzy model the following four rules are used.

Total possible combination of Takagi–Sugeno fuzzy model for TICTL is given by following equation:

$$Total\ possible\ combination\ of\ Takagi-Sugeno\ fuzzy\ model = (No.of linguisticvariable)^{No.of input}$$
$$2^4 = 16 \tag{28}$$

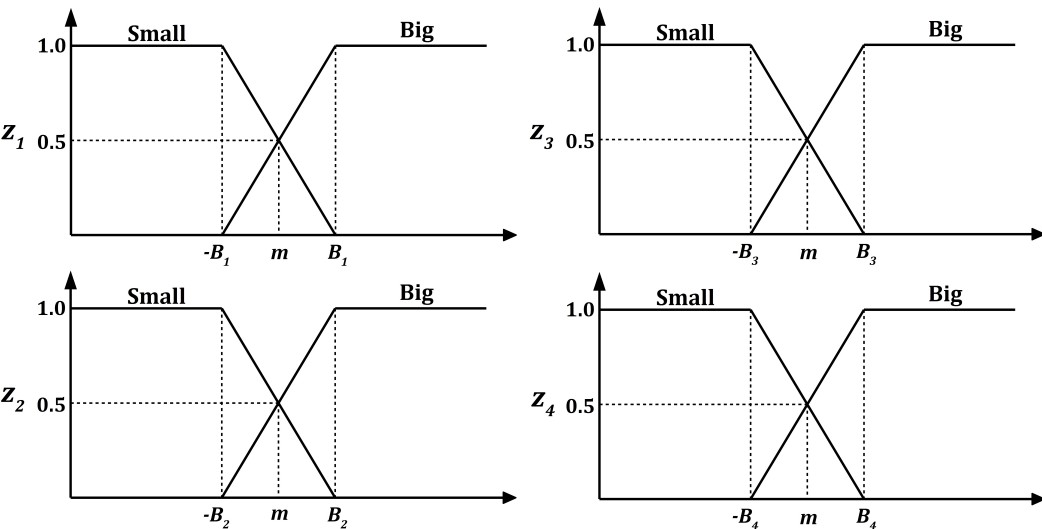

**Figure 5.** Membership functions for four state variables for the TICTL.

The sixteen-rule T–S fuzzy model of TICTL process is as follows:

**Model Rule 1**. *IF $z_1(t)$ is "B" and $z_2(t)$ is "B" and $z_3(t)$ is "B" and $z_4(t)$ is "B" THEN* $\begin{cases} \dot{x} = A_1x + B_1u \\ y = C_1x \end{cases}$

**Model Rule 2**. *IF $z_1(t)$ is "B" and $z_2(t)$ is "B" and $z_3(t)$ is "B" and $z_4(t)$ is "S" THEN* $\begin{cases} \dot{x} = A_2x + B_2u \\ y = C_2x \end{cases}$

$\vdots$

**Model Rule 7**. *IF $z_1(t)$ is "B" and $z_2(t)$ is "S" and $z_3(t)$ is "S" and $z_4(t)$ is "B" THEN* $\begin{cases} \dot{x} = A_3x + B_3u \\ y = C_3x \end{cases}$

**Model Rule 8**. *IF $z_1(t)$ is "B" and $z_2(t)$ is "S" and $z_3(t)$ is "S" and $z_4(t)$ is "S" THEN* $\begin{cases} \dot{x} = A_4 x + B_4 u \\ y = C_4 x \end{cases}$

$$\vdots$$

**Model Rule 15**. *IF $z_1(t)$ is "S" and $z_2(t)$ is "S" and $z_3(t)$ is "S" and $z_4(t)$ is "B" THEN* $\begin{cases} \dot{x} = A_{15} x + B_{15} u \\ y = C_{15} x \end{cases}$

**Model Rule 16**. *IF $z_1(t)$ is "S" and $z_2(t)$ is "S" and $z_3(t)$ is "S" and $z_4(t)$ is "S" THEN* $\begin{cases} \dot{x} = A_{16} x + B_{16} u \\ y = C_{16} x \end{cases}$

where the "B" denotes the "big" membership function and "S" denotes the "small" membership function of the state variable. This fuzzy model exactly represents the dynamics of the nonlinear system under $11.62 \leq z_1(t) \leq 27.26$ and $11.62 \leq z_2(t) \leq 27.26$, $3 \leq z_3 \leq 5$ and $3 \leq z_4 \leq 5$. It is a operating region 3 for TICTL.

Here the TICTL model are given in the form of A, B, and C metrics,

$$A_1 = \begin{bmatrix} -0.01036 & 0.01036 & 0 \\ 0.01036 & -0.01927 & 0.01194 \\ 0 & 0.01194 & -0.01841 \end{bmatrix}, B_1 = \begin{bmatrix} 49.4794 & 0 \\ 0 & 0 \\ 0 & 49.4794 \end{bmatrix}, C_1 = \begin{bmatrix} 1 & 0 & 0 \\ 0 & 0 & 1 \end{bmatrix}$$

$$A_2 = \begin{bmatrix} -0.01036 & 0.01036 & 0 \\ 0 & -0.01927 & 0.01194 \\ 0 & 0.01194 & -0.01841 \end{bmatrix}, B_2 = \begin{bmatrix} 49.4794 & 0 \\ 0 & 0 \\ 0 & 18.8692 \end{bmatrix}, C_2 = \begin{bmatrix} 1 & 0 & 0 \\ 0 & 0 & 1 \end{bmatrix}$$

$$\vdots$$

$$A_7 = \begin{bmatrix} -0.01036 & 0 & 0 \\ 0 & -0.01927 & 0.01194 \\ 0 & 0.01194 & -0.01841 \end{bmatrix}, B_7 = \begin{bmatrix} 18.8692 & 0 \\ 0 & 0 \\ 0 & 49.4794 \end{bmatrix}, C_7 = \begin{bmatrix} 1 & 0 & 0 \\ 0 & 0 & 1 \end{bmatrix}$$

$$A_8 = \begin{bmatrix} -0.01036 & 0 & 0 \\ 0 & -0.01927 & 0 \\ 0 & 0.01194 & -0.01841 \end{bmatrix}, B_8 = \begin{bmatrix} 18.8692 & 0 \\ 0 & 0 \\ 0 & 18.8692 \end{bmatrix}, C_8 = \begin{bmatrix} 1 & 0 & 0 \\ 0 & 0 & 1 \end{bmatrix}$$

$$\vdots$$

$$A_{15} = \begin{bmatrix} -0.01036 & 0 & 0 \\ 0 & -0.01927 & 0.01194 \\ 0 & 0 & -0.01841 \end{bmatrix}, B_{15} = \begin{bmatrix} 18.8692 & 0 \\ 0 & 0 \\ 0 & 49.4794 \end{bmatrix}, C_{15} = \begin{bmatrix} 1 & 0 & 0 \\ 0 & 0 & 1 \end{bmatrix}$$

$$A_{16} = \begin{bmatrix} -0.01036 & 0 & 0 \\ 0 & -0.01927 & 0 \\ 0 & 0 & -0.01841 \end{bmatrix}, B_{16} = \begin{bmatrix} 18.8692 & 0 \\ 0 & 0 \\ 0 & 18.8692 \end{bmatrix}, C_{16} = \begin{bmatrix} 1 & 0 & 0 \\ 0 & 0 & 1 \end{bmatrix}.$$

For the different values of three state variables of TICTL process in operating region 3, total 16 linearized model of TICTL process are presented. For controlling this model linear controllers are designed using parallel distributed compensation. To contest this problem, in Section 3 we demonstrated a method for designing a Takagi–Sugeno fuzzy model-based controller via LMIs, for every range of known values, $h_1$ and $h_3$, related to the operating region of the system. To solve the LMIs and calculating state feedback controller gains $K_i$ a MATLAB platform is used the code was written in LMI Toolbox. The TICTL system response for region 3, is shown in result section where $K_1$ to $K_{16}$ were determined using the LMIs (25) from Theorem 1, we found the subsequent controller gains:

$$K_1 = \begin{bmatrix} 96.26 & 36.60 & 3.22 \end{bmatrix}, \quad K_2 = \begin{bmatrix} 107.94 & 40.57 & 3.70 \end{bmatrix}$$

$$K_3 = \begin{bmatrix} 102.12 & 37.01 & 3.10 \end{bmatrix}, \quad K_4 = \begin{bmatrix} 104.94 & 37.81 & 2.81 \end{bmatrix}$$

$$K_5 = \begin{bmatrix} 94.63 & 34.28 & 3.66 \end{bmatrix}, \quad K_6 = \begin{bmatrix} 96.82 & 35.59 & 3.54 \end{bmatrix}$$

$$K_7 = \begin{bmatrix} 96.13 & 35.98 & 3.27 \end{bmatrix}, \quad K_8 = \begin{bmatrix} 98.13 & 36.18 & 3.22 \end{bmatrix}$$

$$K_9 = \begin{bmatrix} 98.71 & 36.09 & 3.24 \end{bmatrix}, \quad K_{10} = \begin{bmatrix} 99.03 & 36.42 & 3.36 \end{bmatrix}$$

$$K_{11} = \begin{bmatrix} 92.43 & 34.17 & 3.39 \end{bmatrix}, \quad K_{12} = \begin{bmatrix} 94.02 & 34.18 & 3.09 \end{bmatrix}$$

$$K_{13} = \begin{bmatrix} 91.96 & 33.18 & 3.02 \end{bmatrix}, \quad K_{14} = \begin{bmatrix} 92.97 & 33.48 & 3.12 \end{bmatrix}$$

$$K_{15} = \begin{bmatrix} 90.89 & 32.82 & 2.97 \end{bmatrix}, \quad K_{16} = \begin{bmatrix} 91.19 & 33.95 & 3.18 \end{bmatrix}$$

## 4. Simulation Results

In the proposed research, the TICTL process model was established on the mass balance equation. The open loop data was generated and the operating regions are selected from the input output characteristics. The linear T–S fuzzy state space model was developed for TICTL for operating region 3. Also state feedback linear controller was designed for every linear TICTL process model. To simulate the proposed controller MATLAB platform was used, simulation of regulatory and servo responses of TICTL process was carried out in two phases. One was without fault and the other one was with a system component ($f_{sys}$) and actuator ($f_a$) faults. In the first phase, regulatory and servo response of TICTL process was carried out and presented in Figures 6 and 7. The controller parameters of the T–S fuzzy model-based controller is presented in Table 3. Observing the simulation results of Figures 6 and 7 controller is track the reference height efficiently, smoothly and without overshoot.

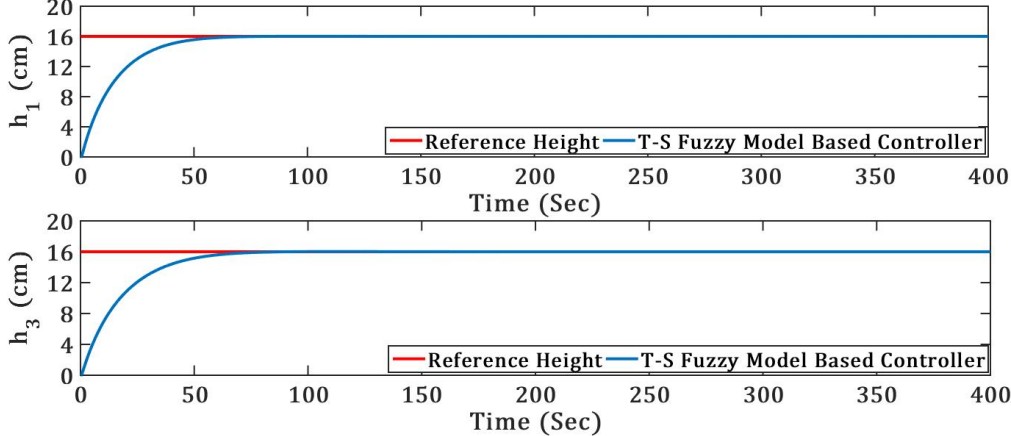

**Figure 6.** Regulatory response of (TICTL) process.

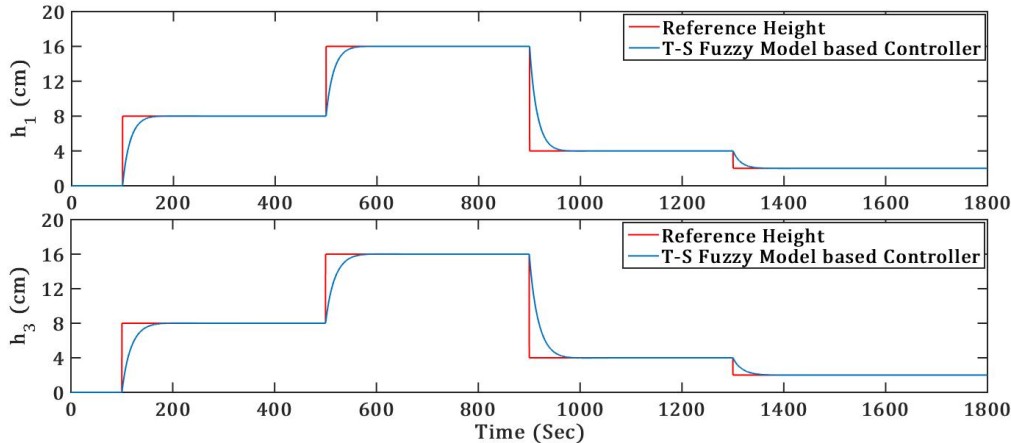

**Figure 7.** Servo response of (TICTL) process.

**Table 3.** Controller performance of T–S fuzzy model-based controller for regulatory response for TICTL process without fault.

| Regulatory Response | Parameters | Parameters for Tank 1 & 3 | |
|---|---|---|---|
| Controller | | $h_1$ | $h_3$ |
| T–S fuzzy Model based Controller | $T_s$ | 54 s | 63 s |
| | $T_r$ | 33.7 s | 40.1 s |
| | $M_p$ (%) | 0 | 0 |

To validate the proposed controller performance against the faulty condition (like system component and actuator faults), it was tested on a prototype model of TICTL with regulatory and servo responses with two possible faults.

Two fault nature is taken for validation (i) Abrupt nature and (ii) Incipient nature.

**Abrupt fault:** The fault occurs into the system at time $t = t_0$ instance and magnitude is constant with respect to time [17]. The abrupt fault behavior with respect to time were modeled by a step function given by Equation (29),

$$f_i\left(t - T_0\right) = \begin{cases} 0 & \text{if } t < T_0 \\ 1 & \text{if } t \geq T_0, \end{cases} \tag{29}$$

where $T_0$ is the occurrence time of the fault.

**Incipient fault:** The fault occurs into the system in between specific time interval and magnitude profile of the fault with respect to time is increasing between time interval [35]. Incipient fault time profiles are modeled by Equation (30),

$$f_i\left(t - T_0\right) = \begin{cases} 0 & \text{if } t < T_0 \\ 1 - e^{-\alpha_i(t-T_0)} & \text{if } t \geq T_0 \end{cases} \tag{30}$$

where the scalar $\alpha_i > 0$ presents the unidentified fault transformation rate. Small values of $\alpha_i$ indicate gradually developing faults, also known as incipient faults [35]. For large values of $\alpha_i$, the time profile $f_i$ approaches a step function, which models abrupt faults.This two nature of system component (leak) and actuator fault was introduced into the TICTL process and tested the proposed controller for regulatory and servo response.

In the second simulation phase regulatory response carried out for TICTL process with two possible fault in abrupt and incipient nature, and found the pre-fault and post-fault response of

proposed controller. To check the controller performance, three integral error indices, namely IAE, ISE and IATE, were calculated for each case. The error formula presented as following equations,

$$IAE = \int |e|\, dt \tag{31}$$

$$ISE = \int \left(e^2\right) dt \tag{32}$$

$$ITAE = \int t\, |e|\, dt, \tag{33}$$

where $e$ is error. First four simulation presented TICTL process regulatory response with possible two nature of faults (system component and actuator faults). As given in the mathematical model, the TICTL process output is $h_1$ and $h_3$. The two output tank height regulatory response is presented in the following subsection. The fault tolerance behavior of the proposed controller was measured with fault recovery time $(T_{fr})$, this terminology given by,

**Fault recovery time ($T_{fr}$):** The time required to achieve previous correct state of the system from faulty state, this terminology is known as fault tolerance ability of the controller.

Proposed T–S fuzzy model-based controller established its effectiveness against the system component (leak) and actuator faults which was proven using fault recovery time results and resulting figures.

### 4.1. Regulatory Response With Faults

(1) TICTL process regulatory with abrupt nature of System component fault

The system component fault considered into this simulation is tank bottom leak in tank 1 and tank 3. At the occurrence of the leak fault into the system drastically level of tank is reduced and hence it degrades the performance drastically, even system will leads to instability when magnitude is big. The two abrupt system component $f_{sys}$ (leak) fault occurs in tank 1 and tank 3 at the same time $t$ = 200 s with fault magnitude M = 5, the regulatory responses of both the tank with system component fault is presented in Figures 8 and 9.

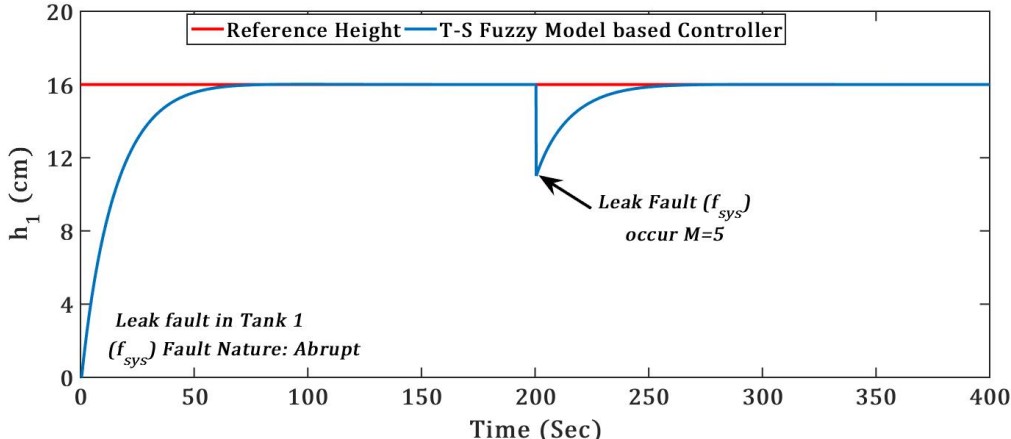

**Figure 8.** Regulatory response of the (TICTL) process with abrupt system component fault in tank 1.

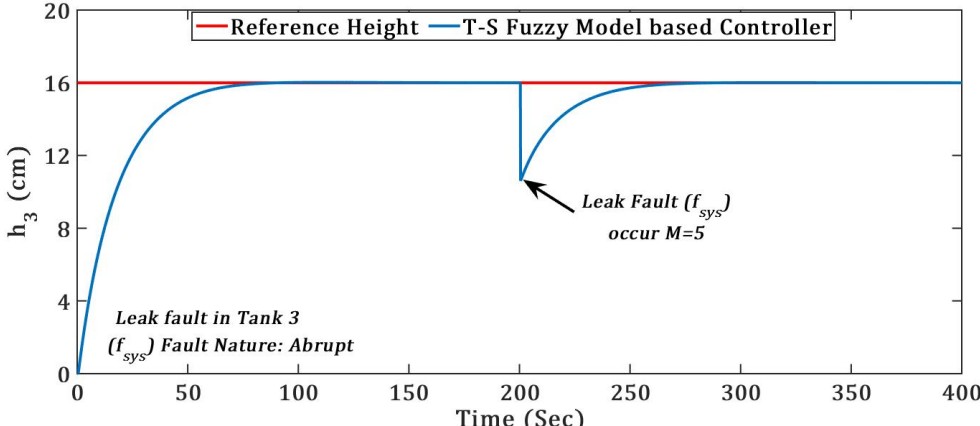

**Figure 9.** Regulatory response of the (TICTL) process with abrupt system component fault in tank 3.

(2)    TICTL process regulatory with incipient nature of System component fault

Now, Figures 10 and 11 presents the regulatory response of the TICTL process with the two incipient system component $f_{sys}$ (leak) fault. The two fault occurred in tanks 1 and 3 separately at the same time $t = 200$ s with fault magnitude M = 5.

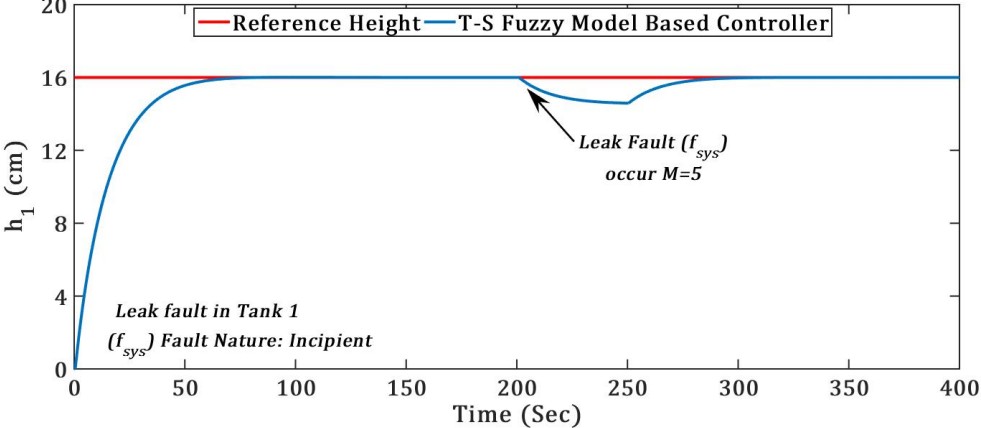

**Figure 10.** Regulatory response of the(TICTL) process with incipient system component fault in tank 1.

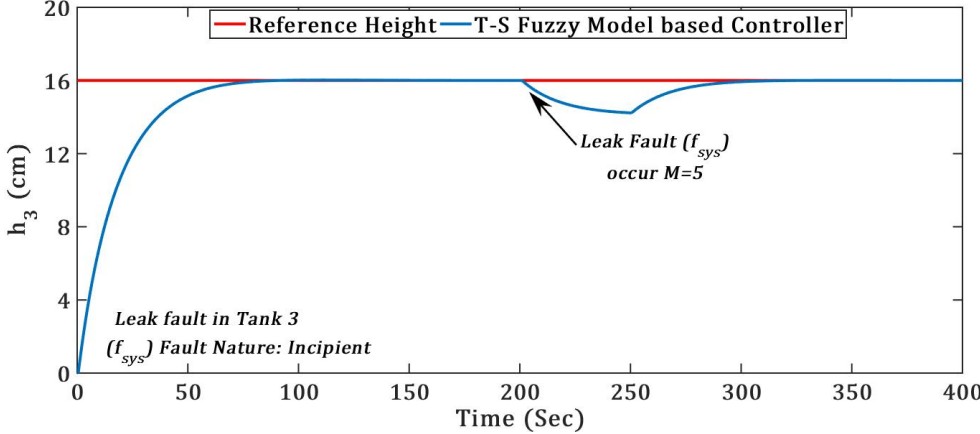

**Figure 11.** Regulatory response of the (TICTL) process with incipient system component fault in tank 3.

(3)    TICTL process regulatory with abrupt nature of actuator fault

An actuator is a major component of any closed loop control system to control the manipulated variable and hence the controlled variable is controlled [36]. Hence, the actuator fault occurring

in the TICTL process conventional controller did not give system stability and optimum response. In Figures 12 and 13, we conferred the stability and optimum response even though the abrupt actuator fault occurred in the TICTL process.

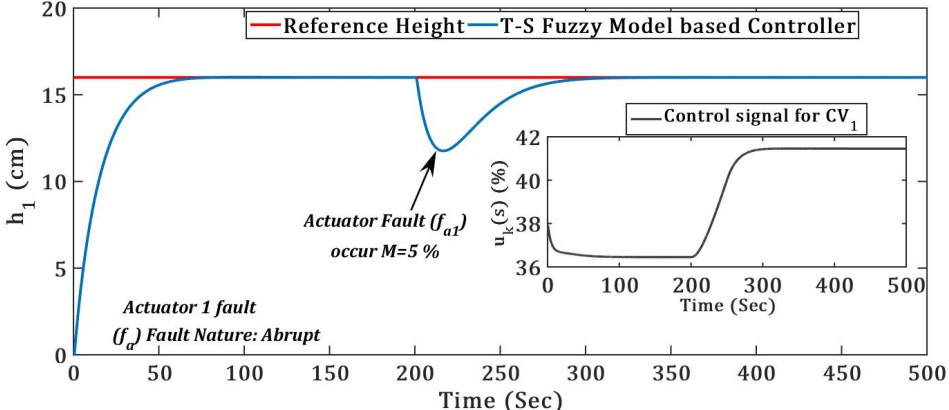

**Figure 12.** Regulatory response of the (TICTL) process with abrupt actuator 1 fault in tank 1.

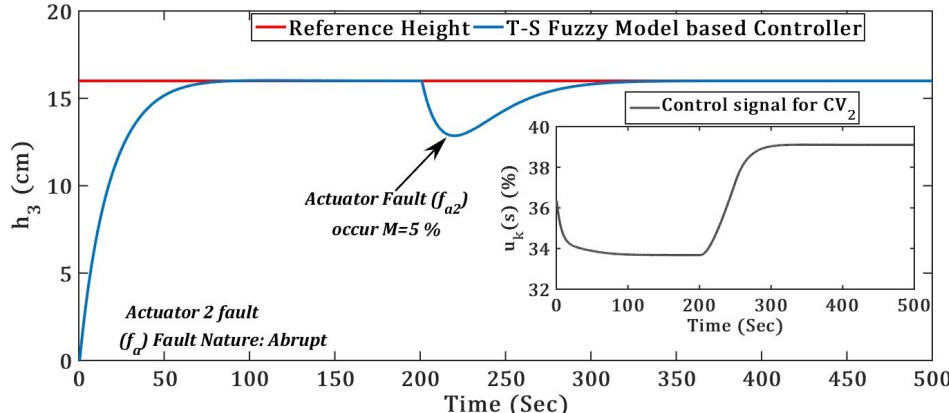

**Figure 13.** Regulatory response of the (TICTL) process with abrupt actuator 2 fault in tank 3.

(4)   TICTL process regulatory with incipient nature of actuator fault

Figures 14 and 15 clearly presents the effectiveness of the proposed T–S fuzzy model-based controller against the incipient actuator fault $(f_{a1})$ and $(f_{a2})$, and fault recovery time $(T_{fr})$ for proposed controller is presented in Table 4. The two separate actuator faults introduced in the TICTL process with same magnitude at time $t$ = 200 s.

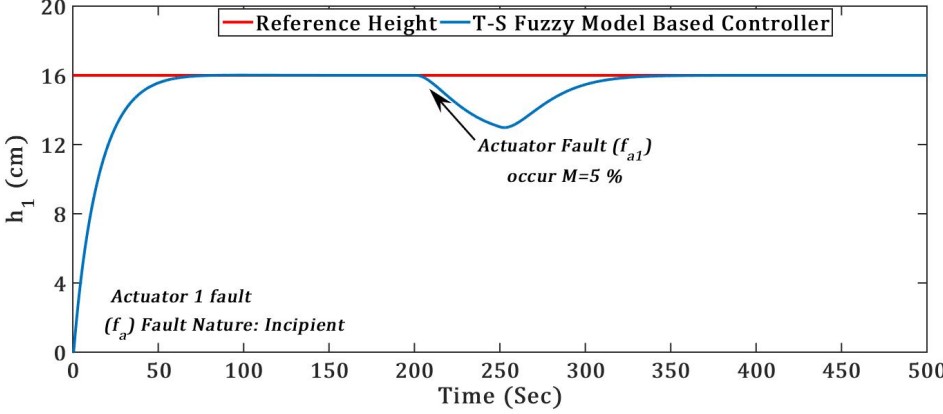

**Figure 14.** Regulatory response of the (TICTL) process with incipient actuator 1 fault in tank 1.

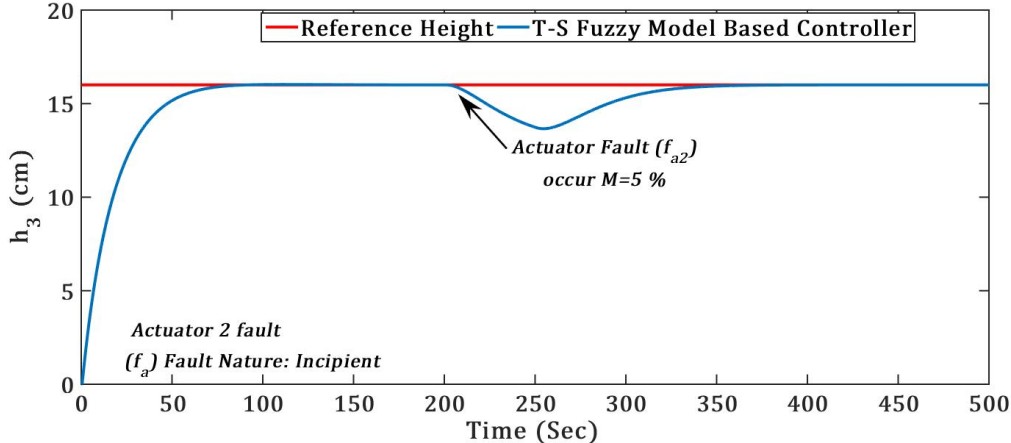

**Figure 15.** Fault-tolerance performance of the T–S fuzzy model-based controller for TICTL subject to two faults.

**Table 4.** Controller Performance for the T–S fuzzy model-based controller for regulatory response for TICTL with fault.

| Regulatory Response | Fault Type with Magnitude | Fault Recovery Time ($T_{fr}$) for Tank 1 & 3 | |
|---|---|---|---|
| **Controller** | | $h_1$ | $h_3$ |
| T–S fuzzy Model based Controller | Leak (Abrupt) (M = 5) | 39 s | 48 s |
| | Leak (Incipient) (M = 5) | 71.5 s | 78.8 s |
| | Actuator (Abrupt) (M = 5%) | 79 s | 90 s |
| | Actuator (Incipient) (M = 5%) | 109 s | 118 s |

System component $f_{sys}$ (Leak) in tank 1 & 3 and Actuator $f_{a1}$ & $f_{a2}$ Faults occur at time $t$ = 200 s.

Steady state behaviour of the controller is presented in Table 5, it is given by Fault Recovery Time ($T_{fr}$) in terms of $T_s$. Integral error indices for proposed controller presented in Table 6 for two type and two nature of faults.

**Table 5.** The T–S fuzzy model-based controller performance in terms of settling time ($T_s$) of the TICTL process subject to different faults in regulatory response.

| Regulatory Response | Fault Type and Magnitude | Fault Recovery Time ($T_{fr}$) in Terms of $T_s$ | |
|---|---|---|---|
| **Controller** | | $h_1$ | $h_3$ |
| T–S fuzzy Model based Controller | Leak (Abrupt) (M = 5) | $0.722 \times T_s$ | $0.7619 \times T_s$ |
| | Leak (Incipient) (M = 5) | $1.3240 \times T_s$ | $1.2507 \times T_s$ |
| | Actuator (Abrupt) (M = 5%) | $1.4629 \times T_s$ | $1.4285 \times T_s$ |
| | Actuator (Incipient) (M = 5%) | $2.0185 \times T_s$ | $1.8730 \times T_s$ |

System component $f_{sys}$ (Leak) in tank 1 & 3 and actuator $f_{a1}$ & $f_{a2}$ Faults occur at time $t$ = 200 s.

**Table 6.** Performance indexes for the T–S fuzzy model-based controller for regulatory response subject to multiple faults.

| Regulatory Response | Fault Type & Nature | IAE | | ISE | | ITAE | |
|---|---|---|---|---|---|---|---|
| **Controller** | | $h_1$ | $h_3$ | $h_1$ | $h_3$ | $h_1$ | $h_3$ |
| T–S fuzzy Model based Controller | Leak (Abrupt) | 0.3681 | 0.4831 | 0.4681 | 0.5439 | 0.6589 | 0.7321 |
| | Leak (Incipient) | 0.6579 | 0.7329 | 0.6891 | 0.7714 | 0.8901 | 0.9541 |
| | Actuator (Abrupt) | 0.8913 | 0.9418 | 0.9602 | 1.0251 | 1.0827 | 1.1891 |
| | Actuator (Incipient) | 1.0398 | 1.1136 | 1.2289 | 1.4101 | 1.3695 | 1.4868 |

Critical Observations for Regulatory Responses of T–S Fuzzy Model-Based Controller

1. From the minute observations of the simulation results, the actuator fault degraded system performance severely when compared with system component (leak) faults.

2. From the two different nature of the faults, the incipient nature of faults will degrade performance significantly as compared to abrupt nature of faults.
3. Fault recovery time $(T_{fr})$ for both the abrupt actuator fault $f_{a1}$ and $f_{a2}$ was almost double as compared to both abrupt system component $f_{sys1}$ and $f_{sys2}$ (leak) faults.
4. Fault recovery time $(T_{fr})$ for both incipient actuator fault $f_{a1}$ and $f_{a2}$ was almost 1.5 times as compared to both incipient system component $f_{sys1}$ and $f_{sys2}$ (leak) faults.
5. Simultaneous fault is introduced into the TICTL process with time, controller simulation and validation were not performed for multiple faults occurs at the same time into the system.

### 4.2. Servo Responses with Multiple Faults in Different Types

Proposed controller efficacy was tested with servo response subject to multiple faults into the TICTL process. Two different natures of leak and actuator faults introduced into the system plant, the simulation results of the proposed controller was presented for tank 1 height ($h_1$) and tank 3 height ($h_3$) are presented in Figures 16 and 17 respectively. Simulation results clearly show the superiority of the controllers irrespective of multiple faults in terms of stability and optimum performance, which was established using three integral error indices depicted in Table 7. Four faults were introduced into the system simultaneously with time, fault 1 $f_{a1}$ occurred at $t$ = 200 s, fault 2 $f_{sys1}$ occurred at $t$ = 600 s, fault 3 $f_{sys2}$ occurred at $t$ = 1100 s, and fault 4 $f_{sys2}$ occurred at $t$ = 1400 s.

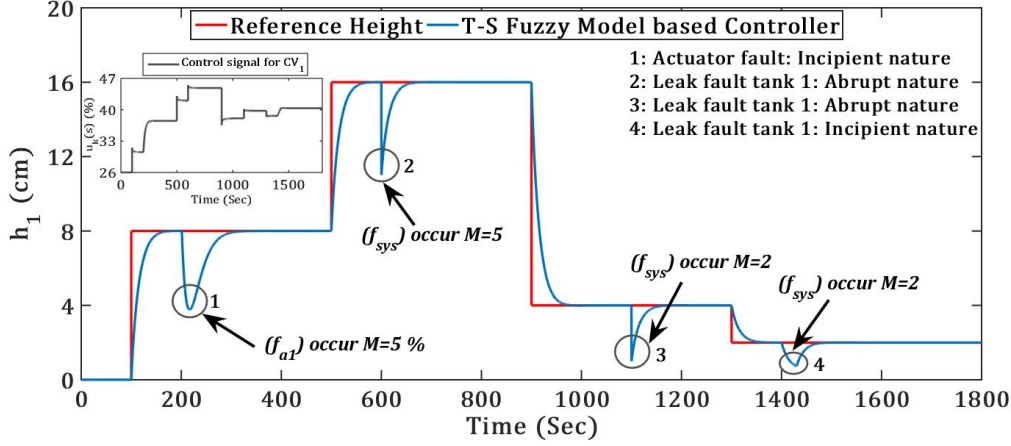

**Figure 16.** Servo response of the (TICTL) process with multiple faults in tank 1.

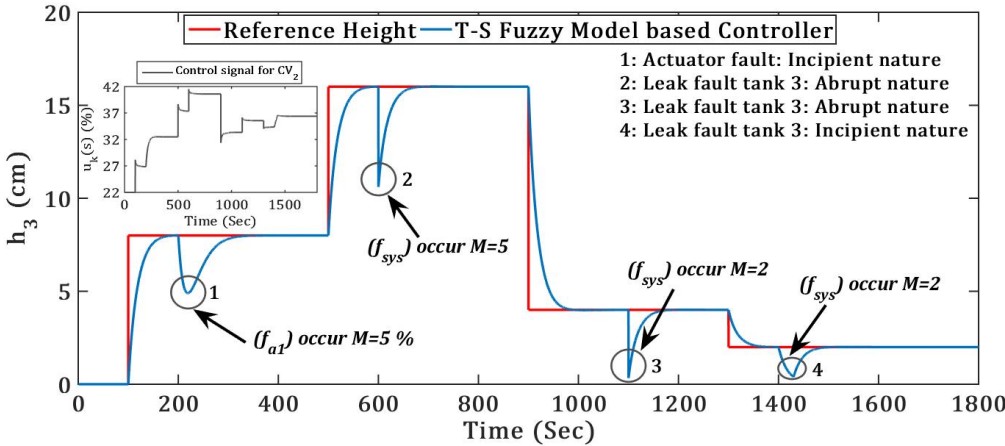

**Figure 17.** Servo response of the (TICTL) process with multiple faults in tank 3.

Critical Observations for Servo Responses of T–S Fuzzy Model-Based Controller

1. The proposed controller tracked the reference height $h_1$ efficiently as compared to reference height $h_3$.

2. Servo response of proposed controller for $h_3$ height gave more to all three integral error indices as compared to servo response of $h_1$ height.

**Table 7.** Performance indexes for T–S fuzzy model-based controller for servo response subject to multiple faults.

| Servo Response | IAE | | ISE | | ITAE | |
|---|---|---|---|---|---|---|
| Controller | $h_1$ | $h_3$ | $h_1$ | $h_3$ | $h_1$ | $h_3$ |
| T–S fuzzy model-based controller | 2.3981 | 3.5328 | 3.1269 | 5.6812 | 5.7612 | 9.3468 |

Every integral performance index had certain advantages in control system design. The ITAE criterion tried to minimize time multiplied absolute error of the control system. The time multiplication term penalized the error more at the later stages than at the beginning and hence effectively reduced the settling time ($t_s$) and percentage of overshoot ($M_p$). At the same time when sudden change of input occurred, the ITAE-based controller design gave lower controller output, hence actuator never saturated and actuator life increased. However the lower controller output was sluggish to the controller response. In Figure 18, the bar chart presents the comparative integral error for regulatory response of TICTL process subject to system component (leak) and actuator faults with abrupt and incipient nature of faults.

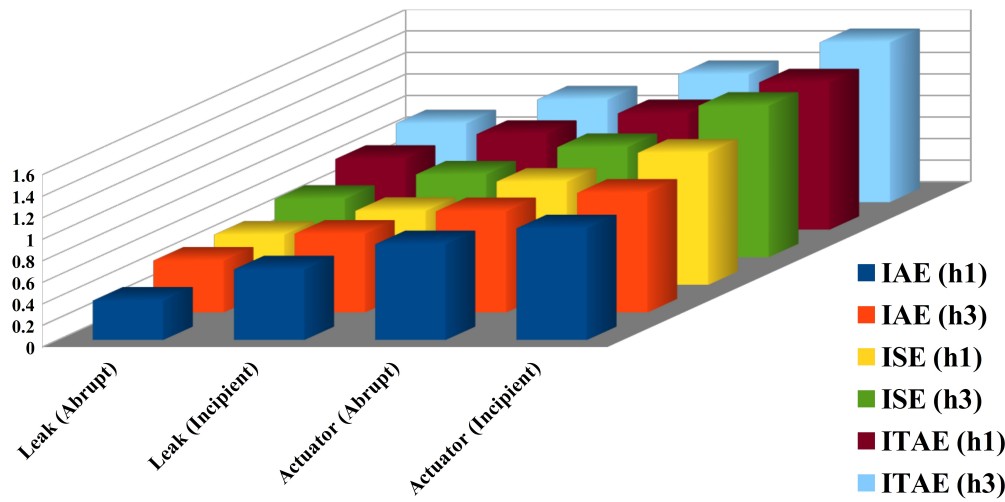

**Figure 18.** Graphical comparison based on integral error indices for two faults and two nature.

## 5. Conclusions

The research article presents methodical design procedure which assures stability and excellent control performance of T–S fuzzy model base control systems. It is discovered that the state feedback controller design for nonlinear system presented by T–S fuzzy models at a local operating region of nonlinear system, can be minimized to a LMIs problem that can be solved using MATLAB software program. Also paper presents three conical tank interacting process is proposed and mathematical model was developed using mass balance equation after that T–S fuzzy model is developed for region 3. It is evident that the present controller design has a superior performance (refer simulation results) even though system component and actuator fault occurs in the the system. Therefore, the specified highly nonlinear system such as TICTL process can be controlled by using this strategy in an uncomplicated, easy, and organized way. The proposed control strategy performances are measured in reference of settling time $t_s$, peak overshoot $M_p$, ISE, IAE, ITAE. Furthermore, the simulation results of the novel fault tolerant controller design for a TICTL process presented with transient and steady state response, as illustrated in the simulation results. Thus, the authors suggest that the proposed strategy can be useful in real-time industrial applications.

As revealed before, the main research scopes in fuzzy control systems are mainly stabilization, optimality and robustness. Recently, these concepts are getting more fascinating due to fuzzy control working superior to conventional control schemes as described in Zadeh's papers [37,38]. In the real world, for any control system the main concern is system stability, according to Zadeh's philosophy the scientific term system's stability are actually fuzzy his one of the paper that in reality most empirical concepts such as stability are actually fuzzy and as demonstration he considers the definition of the Lyapunov stability which is homologous, i.e., a system is either stable or unstable, with no degrees of stability allowed [2]. He identified to the need for reformulation of many basic abstraction in empirical theories with demonstration [2] also Dubois present legacy of the fuzzy logic since inception, and discussed three potential understanding of the grades of membership to a fuzzy set [39].

In the future scope of work, the proposed system is highly nonlinear and modeling uncertainty due to this reason, instead of type 1 fuzzy set (T1FS) interval type 2 fuzzy sets (IT2FS) will be used. The main reason of the using type 2 fuzzy systems for nonlinear and interacting systems is, type 2 fuzzy sets having inherent footprint of uncertainty (FoU) in between upper and lower membership functions which can laminate the model uncertainty as well as giving robustness against the noise [40]. Furthermore, the relaxed stability condition is proposed for type 2 T–S fuzzy model-based controller based on the grade of membership functions and validate it on a proposed system subject to faults and model uncertainty. Also, operating regions 1 and 2 of TICTL process will be the scope of the work. At the same time experimental validation of the proposed controller is the future scope of this current work. The major challenges to implementing IT2FLS is computational time for converting type 2 to type 1 fuzzy sets in the processing unit [41]. But the same advanced algorithm is implemented for fast computation of type 2 to type 1 fuzzy set conversion presented in [42,43].

**Author Contributions:** Conceptualization, H.R.P. and V.A.S.; methodology, H.R.P.; software, H.R.P.; validation, H.R.P.; formal analysis, H.R.P.; investigation, H.R.P.; resources, H.R.P. and V.A.S.; data curation, H.R.P.; writing—original draft preparation, H.R.P.; writing—review and editing, H.R.P. and V.A.S.; supervision, V.A.S.

**Funding:** This research received no external and University funding.

**Acknowledgments:** This paper is an outcome of the ongoing research work of author-1 carrying out a Ph.D. as a part time research scholar at the Dharmsinh Desai University—Nadiad. I would like to express my very great appreciation to Jayesh Barve, M.S. Rao and Jalesh Purohit for his valuable and constructive suggestions during the planning and development of this research work. I also give thanks to Jignesh B. Patel for his constructive technical suggestion for developing the mathematical model of the system and motivation for the development of this research work.

**Conflicts of Interest:** The authors declare no conflict of interest.

## Abbreviations

The following abbreviations are used in this manuscript:

| | |
|---|---|
| MIMO | Multi input multi output |
| SISO | Single input single output |
| FTC | Fault tolerant control |
| IAE | Integral absolute error |
| ITAE | Integral of Time-weighted Absolute Error |
| ISE | Integral square error |
| ISTE | Integral of time-weighted absolute error |
| IT2FLS | Interval type 2 fuzzy logic system |
| IT2FS | Interval type 2 fuzzy sets |
| T1FS | Type 1 fuzzy sets |
| TICTL | Two interacting conical tank level |
| LMI | Linear matrix inequality |
| PDC | Parallel distributed compensation |
| PID | Proportional integral derivative |
| TITO | Two input two output |

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
