# Peer review of "Stable Fault Tolerant Controller Design for Takagi–Sugeno Fuzzy Model-Based Control Systems via Linear Matrix Inequalities: Three Conical Tank Case Study"

_energies, doi:10.3390/en12112221_

Round 1
Reviewer 1 Report
This paper introduces a design algorithm to use the Takagi-Sugeno fuzzy model to simulate a nonlinear control system by mapping that into a series of linear equations that can be solved by numerical software such as MATLAB. The design procedure is illustrated by using the Three Interacting Conical Tank Level (TICTL) model, implemented as a series of mass balance equations, as an example. The control strategy is shown to be able to recover various faults introduced into this control system. This is a well-presented article. I have several minor comments:
1. Line 215: It is better to explain clearly how system component faults and actuator faults were implemented mathematically, and their differences. Also, it is not very clear how the faults were implemented in the context of the previous mathematical formalism.
2. Line 230: How was the error e computed?
3. Line 269: It is scattered in the text, but can the authors explicitly explain the reason for different performance (for example, fault recovery time) between Tank 1 and Tank 3?
4. Line 279: It is not very clear what the authors mean for Point Five.
5. Line 316: There are several grammatical errors in this paragraph that the authors should consider correcting.
Author Response
Respected sir,
Please find attached the reviewer comments replay by the author.

Reviewer 2 Report
Nice piece of work, it is well presented and edited. The list of references definitely needs to be updated. given the fact that this is a highly technical article, you should update the reference list and include more recent (and relevant) articles to support your work. As it stands now there are only 10/28 that come from the last 5-6 years. You need to include more modern research work and you might consider articles like, "A Type of Radial Basis Function Technique for Control and Time Series Prediction of Positioning Systems', Journal of WSEAS Transactions on Systems, ISSN: 2224-2678, Issue 11, Vol. 12, November 2013.
Author Response
Respected sir,
Please find attached response for reviewer comments.
.

Reviewer 3 Report
LMI based control system depsign method is presented for nonlinear system.
1. The controlled plants is too small but the system response is too late such that
I can not find out some interesting point about this paper.
2. Late response may not make interesting problems and issues about controlled system.
More hard conditions on the plants should be proposed.
3. In the results, any comparison results are not presented.
4. In the system responses, the contol inputs should be shown.
5. The authors designed many controllers. We can not see how they work.
Author Response
Respected sir,
Please find attached response to reviewer comments.

Reviewer 4 Report
A fault-tolerant controller based on the Takagi-Sugeno fuzzy model for nonlinear systems is proposed.
The authors should highlight well in the introductory section the limits of the current fuzzy controllers for non-linear systems and the added value of the proposed approach, specifying what are the characteristics of recent approaches in the management literature of non-linear systems that are improved by the proposed model.
In section 3 it is necessary a figure that schematizes the architecture of the proposed model and the partitioning in its processes.
What are the limits of the proposed method? For which nonlinear control systems would its application not be of high performance and the proposed method could be improved?Author Response
Respected sir,
Please find attached the response of reviewer comments.

Round 2
Reviewer 3 Report
I hope that more improved novelty should be presented.
Reviewer 4 Report
In this reviewed paper the authors take into account all my suggestions. I consider this paper publishable in the present form.